# Dynamic metabolic modeling of *Streptomyces clavuligerus* in complex medium highlights nutrient-dependent metabolic transitions associated with clavulanic acid biosynthesis

Leon F. Toro-Navarro[1]*, Laura Pinilla-Mendoza[2], Rigoberto Ríos-Estepa[3]

**1** Escuela de Microbiología, Grupo de Bioprocesos, Universidad de Antioquia, Medellín, Colombia, **2** Departamento de Ingeniería Química, Grupo de Bioprocesos, Universidad de Antioquia, Medellín, Colombia, **3** Grupo de Investigación en Simulación, Diseño, Control y Optimización de Procesos (SIDCOP), Departamento de Ingeniería Química, Universidad de Antioquia, Medellín, Colombia

\* lfelipe.toro@udea.edu.co

## Abstract

### Background

*Streptomyces clavuligerus* is the main industrial producer of clavulanic acid (CA), a potent β-lactamase inhibitor. However, the metabolic interplay linking amino acid utilization, nitrogen regulation, and CA biosynthesis remains poorly understood, particularly under complex medium conditions.

### Methods

The genome-scale metabolic model *iLT1021* was used to simulate the dynamic metabolism of *S. clavuligerus* cultivated in GLYCAS-5 medium. Dynamic flux balance analysis (dFBA) captured the transition from primary to secondary metabolism, while robustness analysis and amino acid network topology identified key metabolic control nodes.

### Results

Integration of dFBA with experimental exometabolome data revealed a clear temporal pattern of amino acid consumption: glutamate, aspartate, and serine were rapidly depleted during exponential growth, whereas histidine and tryptophan were consumed later, coinciding with the onset of CA biosynthesis. Glutamate and glutamine emerged as central nitrogen carriers connecting the α-ketoglutarate and Arg–Orn pathways. Robustness analysis indicated that Arg and Orn enhanced CA fluxes, while Lys and Val had inhibitory effects. Under nitrogen limitation, accumulation of fructose-1,6-bisphosphate and reduced TCA activity reflected a redirection of carbon toward CA precursors.

**Data availability statement:** The amino acid interaction network reconstructed from the iLT1021 genome-scale metabolic model is publicly available in the Network Data Exchange (NDEx) repository under accession ID 4b9fe51c-e7f1-11f0-a218-005056ae3c32. All other relevant data are within the manuscript and its Supporting information files.

**Funding:** The authors acknowledge the support of the Committee for Research Development (CODI) at the University of Antioquia (Grant No. 2022-50930). The funders had no role in study design, data collection and analysis, decision to publish, or preparation of the manuscript.

**Competing interests:** The authors have declared that no competing interests exist.

## Conclusions

This study provides the first genome-scale dynamic modeling of *S. clavuligerus* in a complex medium, demonstrating that amino acid availability and their temporal utilization pattern govern the metabolic switch from growth to CA biosynthesis. This integrative framework helps to interpret the metabolic behavior of *S. clavuligerus* growth in complex medium, offering valuable insights into nutrient regulation and secondary metabolism in filamentous bacteria.

## Introduction

Clavulanic acid (CA) is a molecule with potent β-lactamase inhibitory activity and has become an important commercial product, available both as a generic drug and in combination with β-lactam antibiotics such as amoxicillin and tetracycline [1]. Extensive studies have been conducted to improve antibiotic titers through fermentation processes using *Streptomyces clavuligerus* (*S. clavuligerus*) [1,2]. The productivity of the process depends on several factors, including nutrient type and concentration, operational culture conditions, and intracellular biosynthetic mechanisms [1,3–5].

In industrial fermentations, microorganisms are typically cultivated in nutrient-rich complex media. Under these conditions, bacteria exhibit carbon catabolite repression, preferentially consuming carbon sources that support the highest growth rates [6]. As the microorganism switches from one carbon source to another, growth is temporarily halted while enzymes responsible for metabolizing the new substrates are synthesized. Cellular regulation dynamically adjusts intracellular fluxes across the entire metabolic network to optimize the growth rate [7]. Consequently, regardless of the carbon source used during a specific growth phase, the microbial metabolic network is continuously adjusted, leading to discontinuous changes in metabolic flux distributions as the composition of the culture medium varies [2,8].

A systematic interpretation of such complex behaviors can be achieved through computational modeling with Genome-Scale Metabolic Models (GEMs) [2,9–13]. While *in silico* simulations have advanced to handle multiple substrates in defined media, a significant gap remains in modeling the dynamic and interactive effects of complex, undefined media on metabolic flux distributions. These media contain mixtures of carbon, nitrogen, and other nutrients that are consumed sequentially and interactively, leading to regulatory and metabolic responses not fully captured by simple multi-substrate models [13–15]. Given that complex media exhibit dynamic variations in nutrient availability, integrating GEMs into such studies is crucial for a precise analysis of metabolic effects during CA production.

In this context, Flux Balance Analysis (FBA) and Dynamic FBA (dFBA) have become essential for metabolic analysis [10,13,15–17]. FBA is a steady-state approach that predicts metabolic flux distributions by optimizing an objective function, such as biomass maximization, under fixed environmental conditions [18]. In contrast, dFBA extends this framework by incorporating time-dependent changes in extracellular metabolites. It iterates through discrete time steps, updating nutrient

concentrations and biomass based on uptake and secretion rates, making it particularly suitable for modeling batch and fed-batch cultures [16].

The application of these techniques to the genus *Streptomyces* has grown, with several GEMs now available and providing valuable insights [4,10,15,17,19]. For *S. lividans*, a model has been applied to map extracellular dynamics in a nutritious medium, providing insights for recombinant protein production [9]. This model, designed for recombinant protein production, unraveled the interaction between heterologous protein expression and intracellular metabolism in *S. lividans*, offering novel insights into the influence of cultural and environmental conditions on bioprocess optimization [9]. Another relevant study employed a dynamical modeling approach to investigate secondary metabolism and metabolic switches in *S. xiamenensis* 318 [19]. Through an integrated analysis, combining kinetic metabolic modeling, transcriptase measurements, and metabolic profiling, the study demonstrated that secondary metabolites enhance metabolic fitness by stabilizing the underlying metabolic network. Additionally, fluxes directed toward NADH, NADPH, acetyl-CoA, and glutamate were identified as key metabolic switches governing both primary and secondary metabolism, thus providing valuable insights into optimizing xiamenmycin production [19].

Despite these advancements, a critical application remains unexplored. While the existing studies demonstrate the value of GEMs, they have primarily focused on defined media, different secondary metabolites, or species other than *S. clavuligerus*. The application of dynamic simulation techniques like dFBA to model *S. clavuligerus* cultivated specifically in complex media for CA production is still lacking [10,20].

To address this gap, this study aimed to elucidate the nutritional effects of a complex medium on the growth and metabolism of *S. clavuligerus*. We first applied a genome-scale metabolic model to assess the impact of different nitrogen sources on growth and CA synthesis. Subsequently, we integrated a Monod-type kinetic formulation with dFBA simulations constrained by exometabolome data obtained from batch cultures in a complex medium. This combined computational–experimental framework enables the dynamic characterization of nutrient-dependent metabolic behavior in *S. clavuligerus*, providing a systems-level understanding of how carbon and nitrogen utilization shape the transition from primary to secondary metabolism.

## Materials and methods

### Strain and growth cultures

*S. clavuligerus* ATCC 27064 was used throughout this work. The strain was maintained in 1.5 mL microcentrifuge tubes with sterile glycerol (20% v/v), at −80°C. The content of a single tube was inoculated into 250 mL baffled Erlenmeyer flasks containing 50 mL of TSB medium (seed medium) at pH 7 and incubated for 36h, at 220 rpm and 28°C. Cells from one flask were harvested by centrifugation (10 min at 10,000 rpm). The pellets were resuspended and used as the inoculum in the proposed semi-synthetic medium, GLYCAS-5 (GLYcerol, CAsamino acids and Salts) composed of (g/L): glycerol, 20.0; casamino acids (Caisson Labs®), 5.0; $K_2HPO_4$, 2.0; $CaCl_2$, 0.40; $MnCl_2 \cdot 4H_2O$, 0.10; $FeCl_3 \cdot 6H_2O$, 0.1; $ZnCl_2$, 0.05; $MgSO_4 \cdot 7H_2O$, 1.0; MOPS, 21.0, and the pH was adjusted at 6.8. Cultures were incubated at 28 °C for 140 h at 220 rpm in a rotary shaker. The inoculum consisted of 10% (v/v) seed medium harvested at 36 h. All experiments were performed in triplicate [21].

### Analytical techniques

Clavulanic acid concentration was determined by HPLC (Agilent Technologies Series 1200, Agilent Technologies, Waldbrom, Germany), equipped with a Diode Array Detector DAD (Agilent Technologies, Palo Alto, CA, USA) and an analytical column of 5μm, ZORBAX Eclipse XDB-C18, 4.6 x 150 mm. The mobile phase consisted of a solution of 94% (v/v) of $KH_2PO_4$ (50 mM, pH 3.2) and 6% (v/v) methanol, and the flow rate was 1 mL/min. CA was derivatized with imidazole reagent in a ratio of 1:3 during 30 min at 30 °C. The wavelength of the detector was set at 311 nm [22]. Amino acids were

analyzed by HPLC (Agilent Technologies Series 1200), with a UV/VIS detector and diode array DAD, equipped with a 5 µm analytical column ZORBAX Eclipse AAA-C$_{18}$, 4.6 x 150 mm. The gradient elution was carried out using a quaternary pump to mix the mobile phase composed by water, methanol, acetonitrile and NaH$_2$PO$_4$ buffer (40 mM and pH 7.8) (45:45:10, v:v:v), 40 °C and flow of 2.0 mL/min [21]. Samples were derivatized online using ortho-phthalaldehyde (OPA) for primary amino acids and 9-fluorenyl-methyl-chloroformate (FMOC) for secondary amino acids. Primary amino acids were detected at 338 nm, and the secondary (imino acids) at 262 nm. The calibration curve was prepared using a mixture of amino acid standards at concentrations between 90 and 900 pmol/µL [22]. Glycerol concentration was determined by a spectrophotometric method [23]. Ammonium was determined using the colorimetric Nessler technique [24], using NH$_4$Cl as standard and phosphate was quantified by the reaction with molybdivanadophosphoric acid. Cell biomass concentration was determined gravimetrically as cell dry weight (DW). All analytical measurements were performed using samples obtained from three independent biological replicates (n = 3).

## Kinetic modeling

The unstructured kinetic model consisted of a system of eleven ordinary differential equations describing biomass formation, substrate consumption, and clavulanic acid production. Specifically, the model included differential equations for: (i) biomass growth (X), accounting for both specific growth and death; (ii) glycerol consumption; (iii) individual amino acid uptake (seven amino acids consumed during early exponential phase and nine consumed later in growth); (iv) ammonia and phosphate utilization; and (v) clavulanic acid synthesis, degradation, and the onset of production associated with critical substrate thresholds. Biomass formation was described using Monod-type growth rates associated with glycerol and amino acids, whereas substrate consumption and CA production followed mass-balance ODEs.

In total, the model incorporated five constitutive rate expressions to describe growth, substrate uptake, and the initiation of death and product formation events. Parameter estimation was performed using a nonlinear least-squares approach (Levenberg–Marquardt), fitting all state variables simultaneously against experimental time-course data. A detailed description of the full set of equations, step functions, and parameter-fitting procedure is provided in S1 File.

## In silico analysis

**Amino acid network topological analysis.** To perform the topological analysis of the amino-acid metabolic network in *S. clavuligerus*, we used the published genome-scale metabolic model iLT1021 [12], which includes 1021 genes, 1494 reactions, and 1360 metabolites across central carbon metabolism, amino-acid biosynthesis, secondary metabolism, and clavulanic acid production.

From this GEM, we constructed an amino-acid interaction network in three steps. First, each amino acid was defined as a node. Second, edges were created when two amino acids participated in the same metabolic reaction or when they were connected through sequential reactions forming a linear biosynthetic or degradation pathway. To avoid artificially dense connectivity, ubiquitous cofactors such as ATP, ADP, AMP, NAD(H), NADP(H), FAD, CoA, water, and protons were excluded from the network-building step. Third, when multiple consecutive reactions transformed one amino acid into another through intermediates without branching, these steps were collapsed into a single interaction edge.

The resulting network was imported into Cytoscape (Version 3.10.2) [25], and topological metrics—including degree centrality, betweenness centrality, and clustering coefficient—were computed using Network Analyzer. Amino acids with higher centrality values were considered key nodes within *S. clavuligerus* metabolism.

**Robustness analysis (RA).** To evaluate the effect of each nitrogen source on both growth and clavulanic acid (CA) biosynthesis, we performed a Robustness Analysis (RA) [26]. Glycerol was used as the sole carbon source to provide a standardized baseline, as it supports consistent growth and CA production in *S. clavuligerus* fermentations [12]. To ensure comparability, each nitrogen source was supplied individually at a fixed uptake rate of 1 mmol gDW$^{-1}$ h$^{-1}$, a level low enough to guarantee nitrogen limitation while still supporting measurable metabolic fluxes.

During the RA, the uptake rate of the nitrogen source was varied and, at each point, biomass production was first maximized. Since maximizing growth often results in negligible CA formation, a second optimization step was then performed: CA production was maximized while constraining the growth rate to the optimal value obtained in the first step. This two-stage approach allowed us to quantify the maximum CA flux compatible with the growth optimum at each nitrogen availability level. No CA flux constraints were imposed during the initial growth optimization.

The resulting growth–production space was used to compare the phenotypes achievable with each nitrogen source, extending previous analyses that did not systematically evaluate a wide set of organic and inorganic nitrogen sources [2,4,10].

**Two-step dynamic flux balance analysis (dFBA).** Dynamic flux balance analysis (dFBA) was employed to simulate the time-resolved metabolism of *S. clavuligerus* growing in a complex medium. The overarching workflow for integrating time-course exometabolomic data with dFBA to decipher the dynamics of *S. clavuligerus* metabolism in GLYCAS-5 medium is summarized in Fig 1. Our implementation adapted the established static optimization approach (SOA) [27], which iteratively solves FBA problems at discrete time steps.

The simulation was divided into two consecutive phases to reflect the physiology of *S. clavuligerus*. The initial 22-hour period was designated as the growth phase (trophophase), based on experimental observations that clavulanic acid (CA) production is negligible during this time. For this phase, a standard dFBA was run where the objective at each time step was to maximize the biomass growth rate. The time-dependent constraints for the specific uptake rates of glycerol, amino acids, ammonium, and phosphate, denoted as $v_{Glyc}^{approx}(t)$ and $v_{C_i}^{approx}(t)$, and the specific growth rate, $\mu^{approx}(t)$, were directly applied from the kinetic model (Fig 1, Sections 1 & 2).

After 22 hours, a two-step protocol was implemented at each time step to simulate the onset of the production phase (idiophase). This protocol was designed to model a physiological state where metabolism is optimized for growth, and product formation is coupled to this optimal growth state. The first step determined the maximum possible growth rate under the current nutrient constraints by solving an FBA problem to maximize biomass ($\mu$), subject to the stoichiometric constraints of the iLT1021 model and the kinetically-derived uptake rates. The second step calculated the maximum CA production flux achievable while maintaining the growth rate and key exchange fluxes determined in the first step. This involved solving a second FBA problem to maximize the CA production flux ($v_p^{target}$), with the critical additional constraints that the growth rate ($\mu$) and the glycerol uptake rate ($v_s$) were fixed to their optimal values from the first step.

If the kinetic constraints in the first step of the production phase led to an infeasible solution, an iterative relaxation protocol was applied. The lower and upper bounds for the glycerol uptake and growth rates were progressively widened by a fractional amount *d* (see Fig 1, Section 3) until a feasible solution was obtained. This ensured the simulation could proceed robustly even when the kinetic model's point estimates were slightly inconsistent with the stoichiometric constraints of the metabolic network.

All simulations were performed using the COBRA Toolbox [26] in a MATLAB environment.

## Results

### Culture medium and growth kinetics

GLYCAS-5 is a rich semi-synthetic medium formulated with both organic carbon and nitrogen sources. Glycerol serves as the primary carbon source, while casein hydrolysate provides amino acids and peptides that sustain growth and stimulate secondary metabolism. The amino acid composition of the GLYCAS-5 medium is shown in Fig 2. Amino acids from the glutamate and aspartate families were predominant, together accounting for more than half of the total amino acid content. The pyruvate and serine families contributed moderate proportions, whereas aromatic and histidine family members were detected at lower levels.

This composition reveals that GLYCAS-5 contains key amino acids involved in central carbon and nitrogen metabolism, including L-glutamate, L-aspartate, L-leucine, and L-proline. In contrast, L-asparagine, L-glutamine, and methionine were

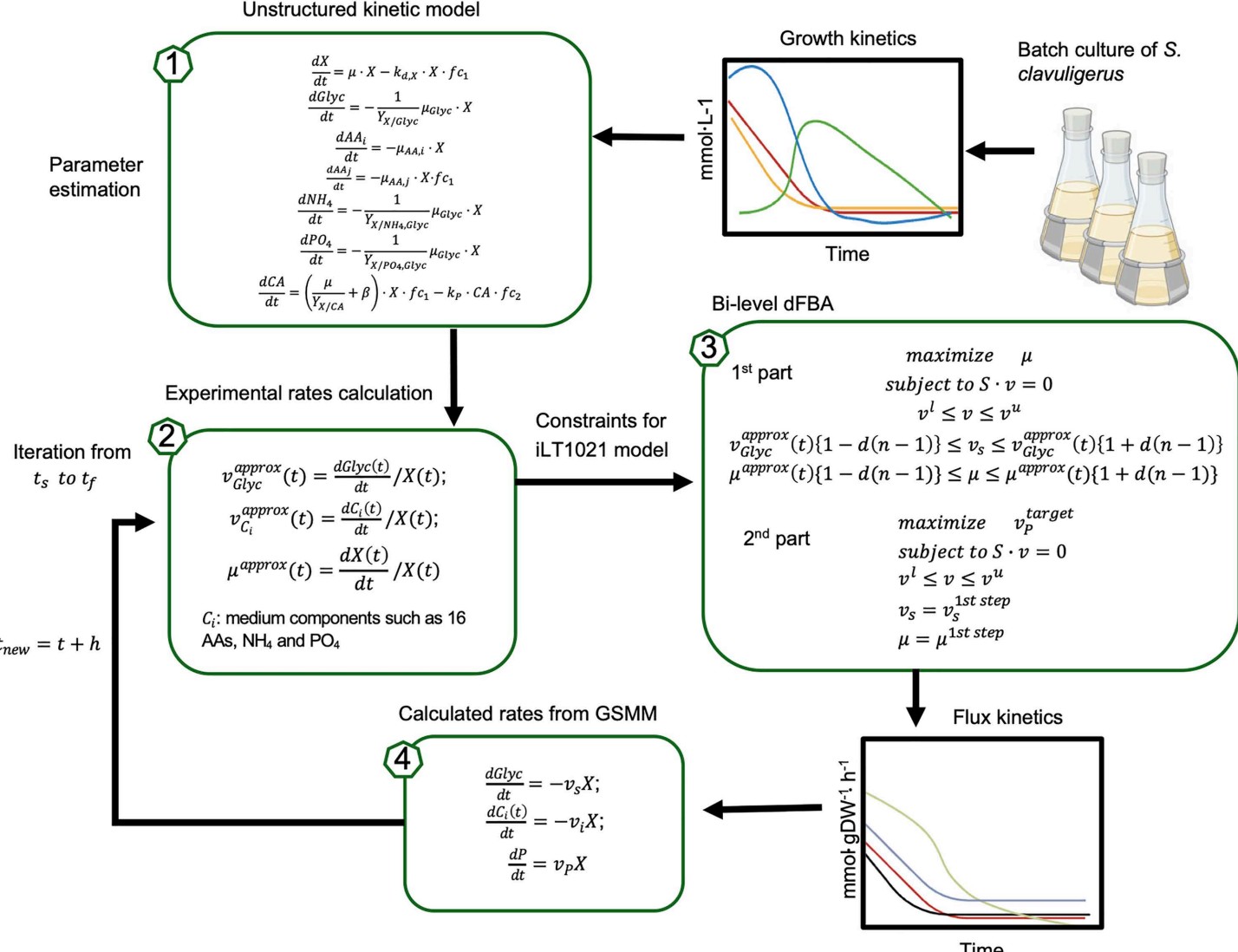

**Fig 1. Flowchart of the dynamic flux balance analysis (dFBA) framework used to describe the metabolic dynamics of *Streptomyces clavuligerus* growing in GLYCAS-5 medium.** The scheme summarizes experimental inputs, model constraints, and simulation outputs integrated in the modeling workflow.

not detected. The balanced representation of amino acid families suggests that GLYCAS-5 provided an effective nutrient framework to support both biomass formation and CA biosynthesis. Although several studies have explored amino acid supplementation strategies for enhancing CA and/or cephamycin C production, none have evaluated the integrated effect of complex nitrogen sources on both secondary metabolism and cell growth [9].

The growth dynamics of *S. clavuligerus* ATCC 27064 in batch cultures are presented in Fig 3. The identified growth phases were: early exponential phase (EEP, 0–20 h), exponential phase (EP, 21–40 h), stationary phase (SP, 41–59 h), and death phase (DP, ≥ 60 h). No lag phase was observed, likely due to the high inoculum biomass from the seed culture. Additionally, the trophophase (TP, growth phase) and idiophase (IP, secondary metabolite production phase) occurred between 0–40 h and 40–71 h, respectively. During the TP, cells grew rapidly and formed pellets, consistent with previously reported morphologies [28]. In contrast, the IP was characterized by slower growth and maximal CA accumulation [2].

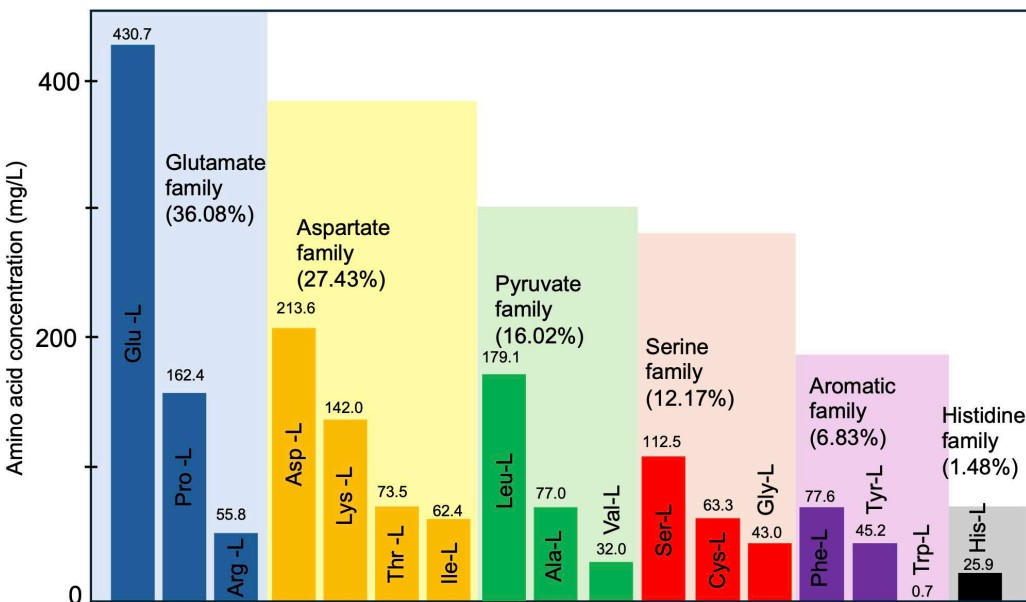

**Fig 2. Amino acid composition of the GLYCAS-5 medium.** Quantitative analysis of amino acid concentrations (mg/L) determined from three independent biological replicates (n = 3). Values are reported as mean concentrations.

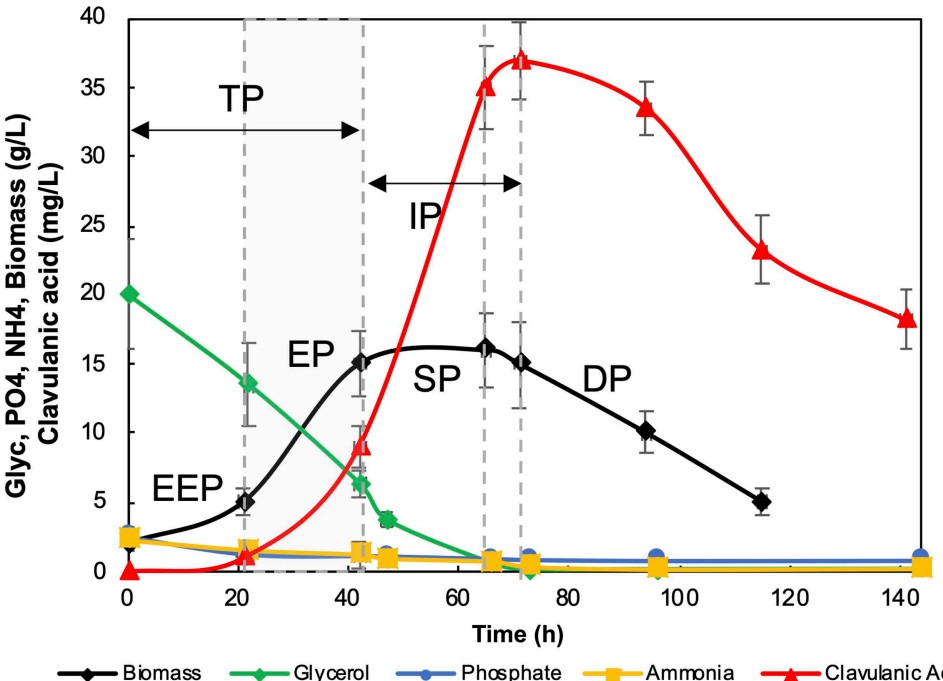

**Fig 3. Growth kinetics of *Streptomyces clavuligerus* ATCC 27064 in the GLYCAS-5 medium.** Concentrations were determined from three independent biological replicates (n = 3). Growth phases are indicated as follows: EEP, early exponential phase; EP, exponential phase (grey zone); SP, stationary phase; DP, death phase; TP, trophophase; IP, idiophase.

Batch cultures in GLYCAS-5 yielded a maximum CA concentration of 36.0 ± 0.52 mg/L at 71 h, coinciding with the end of the IP (Fig 3). Rapid exponential growth led to early nutrient depletion and a subsequent decline in biomass after reaching 16.3 ± 0.34 g/L at 42 h. Glycerol, the main carbon source, was rapidly consumed and completely depleted at 73 h, coinciding with peak CA production and no phosphate limitation was observed.

Other medium components, such as phosphate, were approximately 70% consumed by 70 h, leaving a residual concentration of 0.76 g/L at the end of the culture, indicating that phosphate was not growth-limiting. The accumulation of free ammonium, common in amino acid–rich fermentations, reached 0.15 g/L at 144 h, likely resulting from the catabolic degradation of amino acids [9].

The time-course profiles of amino acid concentrations were experimentally measured and are shown in Fig 4A–4D. A sequential pattern of amino acid depletion was observed during growth. L-aspartate, L-glutamate, and L-serine showed the fastest depletion rates, coinciding with glycerol exhaustion. During the early exponential phase (EEP), transient increases in L-leucine, L-cysteine, L-arginine, L-tyrosine, and L-glycine concentrations were observed. In contrast, L-aspartate, L-serine, L-histidine, and L-phenylalanine were completely consumed during the exponential phase (EP). As cultures entered the stationary phase (SP), L-glycine and L-leucine were further

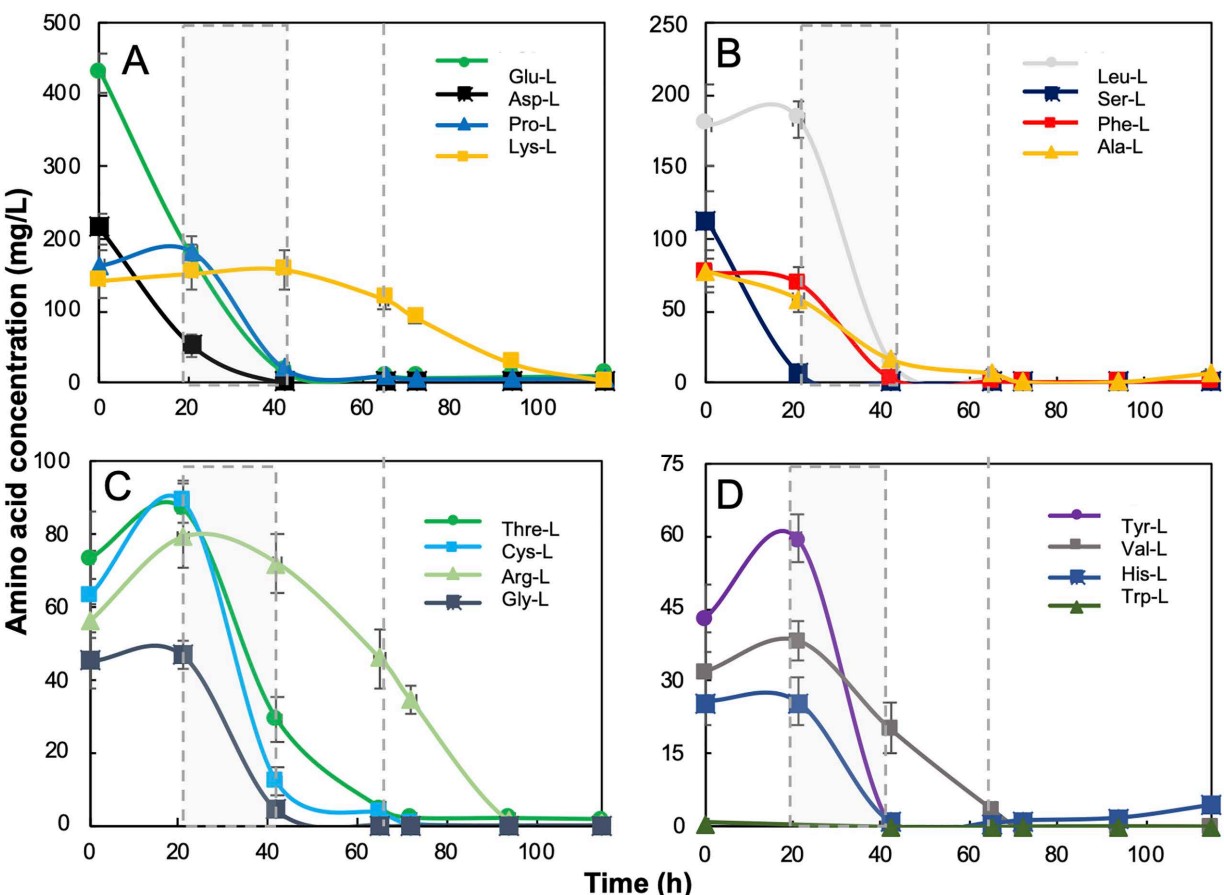

**Fig 4. Dynamics of amino acid consumption by *Streptomyces clavuligerus* ATCC 27064 during growth in GLYCAS-5 medium.** Concentrations were determined from three independent biological replicates (n = 3). **(A)** Glutamate (L-Glu), Aspartate (L-Asp), Proline (L-Pro), and Lysine (L-Lys); **(B)** Leucine (L-Leu), Serine (L-Ser), Phenylalanine (L-Phe), and Alanine (L-Ala); **(C)** Threonine (L-Thr), Cysteine (L-Cys), Arginine (L-Arg), and Glycine (L-Gly); **(D)** Tyrosine (L-Tyr), Valine (L-Val), Histidine (L-His), and Tryptophan (L-Trp).

utilized, and during the death phase (DP), the remaining amino acids were depleted, with L-arginine and L-lysine becoming undetectable after 80 h.

## Unstructured kinetic model of *S. clavuligerus* growing in the GLYCAS-5 medium

The proposed dynamic first-principles model (equations (1)–(11) in S1 File) was formulated considering only nutritional factors in the system of ordinary differential equations (ODEs) [29–32]. The estimated kinetic parameters for the unstructured dynamic model describing *S. clavuligerus* growth and CA production in GLYCAS-5 medium are summarized in Table 1.

The model exhibited excellent agreement between simulations and experimental data, with determination coefficients ($R^2$) of 0.948 for biomass, 0.986 for glycerol, and 0.985 for CA. Amino acids such as L-glutamate, L-aspartate, and L-serine showed near-perfect fits ($R^2 > 0.999$). In contrast, lower correlations were obtained for phosphate ($R^2 = 0.423$) and ammonium ($R^2 = 0.769$). The time-course simulations and the corresponding experimental data used for parameter estimation are provided in S1 File.

The estimated kinetic parameters were consistent with the experimental observations described for the GLYCAS-5 medium. The biomass-yield coefficients reported in Table 1 are expressed in gDW per mmol, consistently with the mmol-based mass balances used in the unstructured dynamic model. These coefficients act as effective fitted parameters within a multi-substrate framework rather than as standalone physiological biomass yields.

Amino acids from the glutamate and aspartate families—predominant in the medium and rapidly consumed during the trophophase—exhibited high specific growth rates ($\mu_{Glu}^{max} = 0.006$ h$^{-1}$, $\mu_{Asp}^{max} = 0.0074$ h$^{-1}$) and high biomass yields ($Y_{X/Glu} = 32.7$ gDW mmol$^{-1}$, $Y_{X/Asp} = 0.843$ gDW mmol$^{-1}$). These results confirm their central role as preferred carbon–nitrogen sources supporting rapid biomass accumulation during the exponential phase. Conversely, amino acids that accumulated transiently in the early exponential phase, such as L-arginine and L-lysine, showed the lowest $\mu_i^{max}$ values (0.002 h$^{-1}$), consistent with their slower assimilation or regulatory control in nitrogen metabolism.

The low affinity estimated for glycerol ($K_{Glyc} = 79.4$ mmol L$^{-1}$) agrees with its progressive depletion observed experimentally, indicating sustained utilization rather than preferential consumption. This aligns with the observed transition from the trophophase to the idiophase, during which glycerol exhaustion coincided with maximal CA accumulation. Overall, the parameter estimates not only captured the global dynamics of biomass and product formation but also reflected the compositional and temporal patterns observed in the GLYCAS-5 medium. This coherence supports the model's ability to quantitatively describe the interplay between carbon and nitrogen metabolism underlying CA biosynthesis.

## In silico analysis

**Topology of amino acid network extracted from iLT1021.** The topology of the amino acid interaction network reconstructed from the iLT1021 genome-scale metabolic model (GEM) is depicted in Fig 5. This network integrates the main biosynthetic routes involving amino acids and identifies key topological features that define the organism's metabolic organization. Major biosynthetic routes, including protein, peptidoglycan, and TBC1H biosynthesis, act as central metabolic hubs connecting primary and secondary pathways. The amino acid interaction network is publicly available through the Network Data Exchange (NDEx) [33].

Among amino acids, L-glutamate (Glu), glycine (Gly), L-cysteine (Cys), and L-aspartate (Asp) emerged as highly connected metabolites, exhibiting strong topological centrality and extensive cross-pathway interactions. In contrast, amino acids such as L-threonine (Thr), L-histidine (His), and L-lysine (Lys) displayed more specialized, pathway-specific roles. As shown in Fig 5A and 5C, Glu presented the highest number of interactions (361) and connected nodes (133), confirming its function as a key hub in nitrogen metabolism and precursor supply. Gly and Ala also displayed broad interconnectivity, linking multiple subsystems, whereas Cys, Asp, and Ser maintained moderate connectivity levels, suggesting complementary but less dominant regulatory roles. Conversely, His, Arg, and Phe showed limited connectivity, consistent with their participation in discrete biosynthetic branches.

**Table 1. Estimated kinetic parameters for the unstructured dynamic model describing *Streptomyces clavuligerus* growth and CA production in GLYCAS-5 medium.**

| Symbol | Value | Symbol | Value |
|---|---|---|---|
| **Maximum specific growth rates, $\mu_i^{max}$, 1/ h** | | | |
| $\mu_{Glyc}^{max}$ | 0.0100 | $\mu_{Glu}^{max}$ | 0.0060 |
| $\mu_{Asp}^{max}$ | 0.0074 | $\mu_{Ser}^{max}$ | 0.0063 |
| $\mu_{Trp}^{max}$ | 0.0990 | $\mu_{Phe}^{max}$ | 0.0073 |
| $\mu_{Ala}^{max}$ | 0.0079 | $\mu_{His}^{max}$ | 0.0031 |
| $\mu_{Pro}^{max}$ | 0.0138 | $\mu_{Thr}^{max}$ | 0.0402 |
| $\mu_{Lys}^{max}$ | 0.0020 | $\mu_{Cys}^{max}$ | 0.1560 |
| $\mu_{Arg}^{max}$ | 0.0020 | $\mu_{Leu}^{max}$ | 0.0080 |
| $\mu_{Tyr}^{max}$ | 0.0115 | $\mu_{Gly}^{max}$ | 0.0096 |
| $\mu_{Val}^{max}$ | 0.0163 | | |
| **Substrate half saturation constant, Ki, mmol/ L** | | | |
| $K_{Glyc}$ | 79.4000 | $K_{Glu}$ | 0.1020 |
| $K_{Asp}$ | $1.15 \times 10^{-8}$ | $K_{Ser}$ | 0.0017 |
| $K_{Trp}$ | 0.0250 | $K_{Phe}$ | $6.60 \times 10^{-9}$ |
| $K_{Ala}$ | 0.00225 | $K_{His}$ | $1.21 \times 10^{-8}$ |
| $K_{Pro}$ | $6.84 \times 10^{-5}$ | $K_{Thr}$ | 1.4900 |
| $K_{Lys}$ | 0.0014 | $K_{Cys}$ | 0.1680 |
| $K_{Arg}$ | $7.59 \times 10^{-8}$ | $K_{Leu}$ | 0.0139 |
| $K_{Tyr}$ | 0.0379 | $K_{Gly}$ | $4.23 \times 10^{-6}$ |
| $K_{Val}$ | $1.55 \times 10^{-4}$ | | |
| **Compound-related biomass yield coefficients, $Y_{X/i}$, gDW/mmol** | | | |
| $Y_{X/Glyc}$ | 0.0204 | $Y_{X/Glu}$ | 32.7000 |
| $Y_{X/Asp}$ | 0.8430 | $Y_{X/Ser}$ | 0.8110 |
| $Y_{X/Trp}$ | 0.1130 | $Y_{X/Phe}$ | 0.0165 |
| $Y_{X/Ala}$ | 14.4000 | $Y_{X/His}$ | $6.29 \times 10^{-5}$ |
| $Y_{X/Pro}$ | $6.05 \times 10^{-9}$ | $Y_{X/Thr}$ | 2.6900 |
| $Y_{X/Lys}$ | 0.2370 | $Y_{X/Cys}$ | 15.9000 |
| $Y_{X/Arg}$ | 0.7130 | $Y_{X/Leu}$ | 0.1340 |
| $Y_{X/Tyr}$ | $2.03 \times 10^{-5}$ | $Y_{X/Gly}$ | 0.2780 |
| $Y_{X/Val}$ | 0.0236 | $Y_{X/PO_4,Glyc}$ | 0.0059 |
| $Y_{X/NH_4,Glyc}$ | 0.0339 | $Y_{X/CA}$ | 158.8000 |
| **Specific death constant, 1/h** | | $k_{d,X}$ | 0.0218 |
| **Clavulanic acid degradation constant, 1/h** | | $k_P$ | 0.0147 |
| **Growth-associated product formation coefficient (mg/ g)** | | $\alpha$ | $1.39 \times 10^{-4}$ |
| $C_{S1}$, g/L | | | 4.860 |
| $C_{S2}$, g/L | | | 24.500 |

The quantitative analysis of network parameters (Fig 5B, 5C) revealed a negative correlation between average shortest path length and centrality measures, indicating that highly connected metabolites shorten overall metabolic distances and enhance flux efficiency. Meanwhile, the topological coefficient exhibited weak correlations with other parameters, implying that some hubs—particularly Glu and Gly—possess unique, non-redundant connectivity patterns that may confer specific regulatory control.

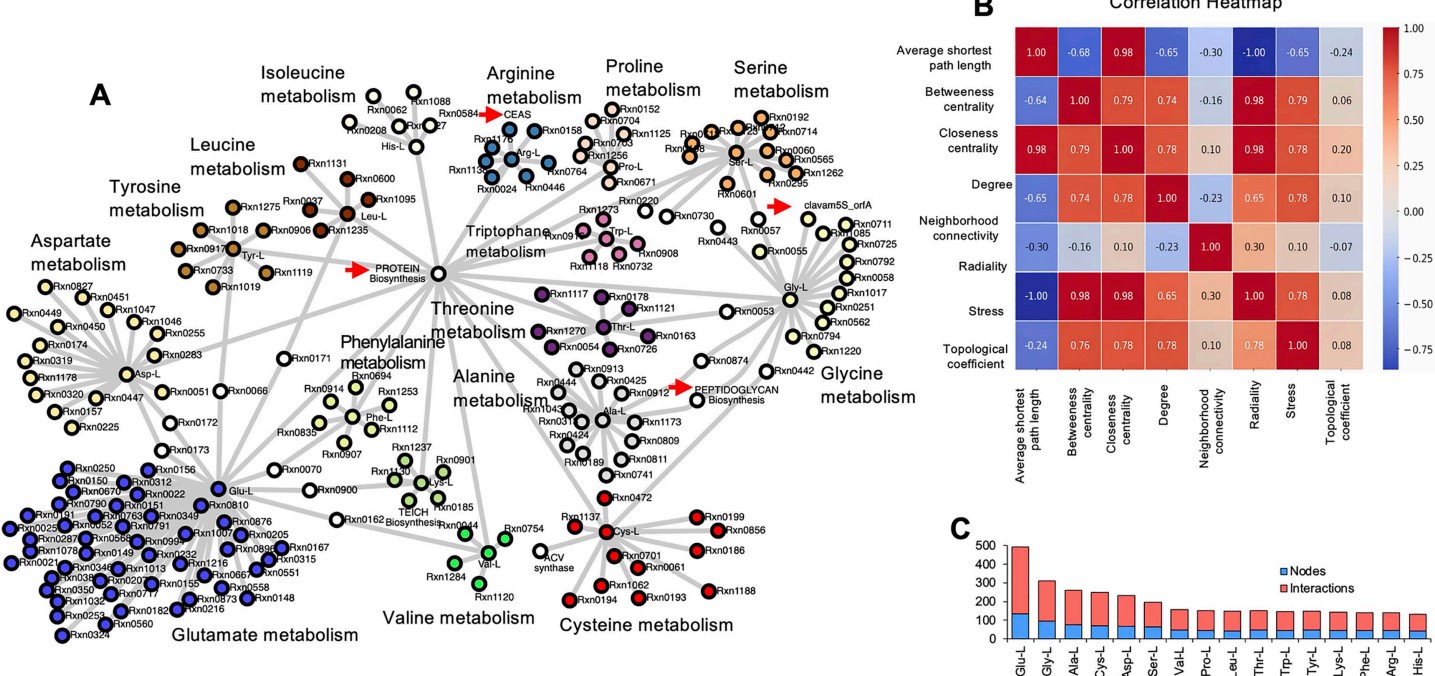

**Fig 5. Topological analysis of the amino acid metabolic network derived from the iLT1021 GEM. (A)** Central hub metabolites and their interaction map, **(B)** Correlation heatmap of network topology parameters, **(C)** Quantitative distribution of connectivity metrics across amino acids.

Altogether, the network topology derived from the iLT1021 model underscores how amino acid metabolism in *S. clavuligerus* is structured around a few multifunctional hubs that interlink carbon and nitrogen metabolism with secondary metabolite biosynthetic routes. These results provide a systems-level perspective that complements the experimental observations in GLYCAS-5 medium and set the stage for analyzing the metabolic shifts leading to CA synthesis.

**Robustness analysis (RA).** To evaluate the influence of different nitrogen sources on growth and CA biosynthesis, the metabolic response of *S. clavuligerus* was analyzed using flux balance analysis (FBA) coupled with robustness analysis (RA). Eighteen nitrogen sources were assessed, including sixteen amino acids and two inorganic alternatives—ammonium and urea (S2 File). Fig 6 summarizes the *in silico* evaluation of nitrogen source utilization and its impact on CA production and biomass formation.

As shown in Fig 6A, *in silico* simulations identified L-arginine (Arg) and L-ornithine (Orn) as the most favorable nitrogen sources for CA production, achieving maximum rates of 0.33 mmol·gDW$^{-1}$·h$^{-1}$. These amino acids also supported significant biomass accumulation, with growth rates of 0.23 h$^{-1}$, representing the optimal balance between CA synthesis and cellular proliferation.

L-glutamine (Gln) also exhibited high efficacy, with production rates comparable to those of Arg and Orn. Amino acids belonging to the α-ketoglutarate family—including L-glutamate (Glu), L-glutamine (Gln), and L-proline (Pro)—were particularly effective (Fig 6B). These nitrogen sources achieved CA production rates near 0.33 mmol·gDW$^{-1}$·h$^{-1}$ while sustaining moderate growth (~0.11 h$^{-1}$), a characteristic advantageous for antibiotic fermentation processes that prioritize metabolite yield over biomass formation [10].

In contrast, amino acids from the oxaloacetate family (Asn, Thr, Lys, Hom) and the 3-phosphoglycerate family (Cys, Ser, Gly) supported only moderate CA biosynthesis, with production rates ranging from 0.23 to 0.27 mmol·gDW$^{-1}$·h$^{-1}$

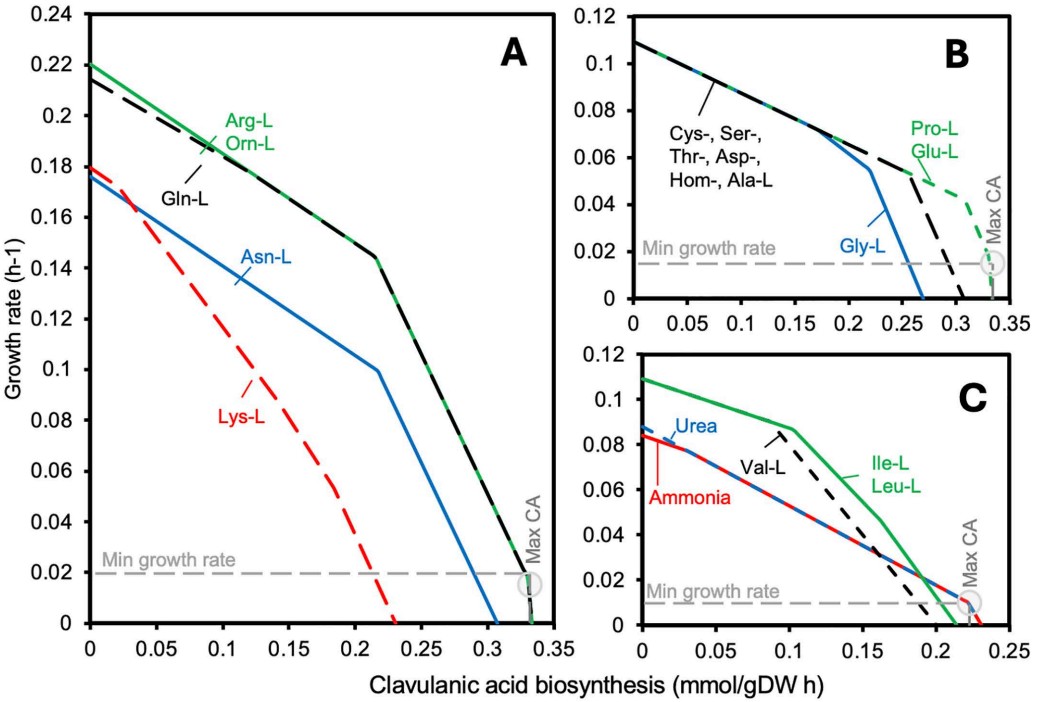

**Fig 6. Nitrogen source effect on clavulanic acid biosynthesis and growth using glycerol as a sole carbon source (1 mmol/gDW·h).** The intersection of different growth profiles, on specific amino acids, corresponds to the best scenario for CA synthesis while maintaining a minimum *Streptomyces clavuligerus* biomass growth.

(Fig 6A and 6B). Similarly, L-alanine (Ala) from the pyruvate family displayed rates within this range but was less effective than the optimal α-ketoglutarate-derived sources.

Among all nitrogen sources evaluated, L-valine (Val) showed the lowest CA production ($<0.20$ mmol·gDW$^{-1}$·h$^{-1}$) and growth rates below 0.13 h$^{-1}$, while Ala, Lys, and His also supported minimal CA accumulation, confirming their limited contribution to secondary metabolism (Fig 6C) [3].

Finally, ammonium ($NH_4^+$) and urea sustained only low growth rates and negligible CA production (Fig 6C). These results emphasize that amino acid-derived nitrogen sources are essential for optimizing CA biosynthesis in *S. clavuligerus*. Overall, L-Arg and L-Orn emerged as the most effective nitrogen sources, providing an optimal balance between high CA yields and sufficient biomass formation, in agreement with previous reports [34,35].

**Dynamic flux balance analysis.** Fig 7 shows the dFBA simulations for *S. clavuligerus* growth in the GLYCAS-5 medium, complete metabolic flux distribution is included in S2 File. A comparison between simulation results and experimental data is presented in S3 File.

During the early exponential phase (EEP) and the exponential phase (EP), glycerol was rapidly consumed, with its uptake rate declining significantly after 20 hours, while ammonium exhibited a similar consumption pattern, highlighting its role as a key nutrient during the initial growth stages (Fig 7A). The growth rate remained high during the first 20 hours (~0.17 h$^{-1}$) and decreased thereafter, coinciding with the depletion of glycerol and essential amino acids required for biomass synthesis. Phosphate uptake was initially constant but gradually decreased over time, indicating progressive consumption throughout the culture period.

Oxygen uptake was elevated during the first 20 hours, reflecting intense metabolic activity and high respiratory demand, while $CO_2$ secretion followed a comparable pattern, peaking during EEP and EP and declining after 20 hours

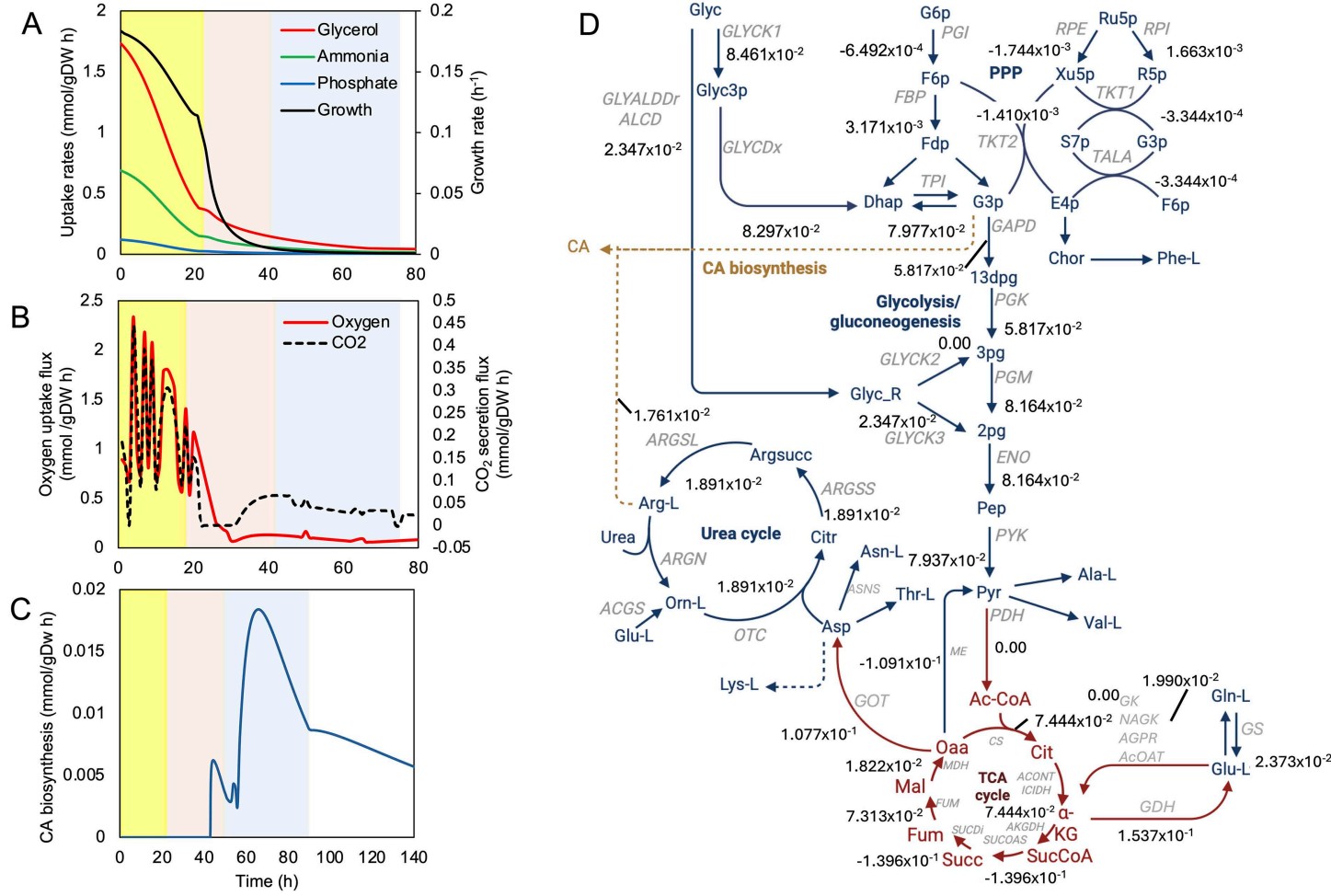

**Fig 7. Dynamic flux balance analysis (dFBA) of _S. clavuligerus_ in GLYCAS-5 medium. (A)** Metabolic fluxes during early exponential (yellow), stationary (red), and decay (blue) phases. Phases are demarcated by glycerol depletion and a decrease in growth rate. **(B)** Oxygen uptake and $CO_2$ secretion fluxes. **(C)** CA secretion flux. **(D)** Carbon flux distribution contributing to CA biosynthesis in decay phase (DP).

(Fig 7B). CA secretion commenced at approximately 40 hours, coinciding with the transition from exponential growth to the stationary phase (SP). The secretion flux of CA reached its maximum around 70 hours, corresponding to the early SP, and gradually declined after 80 hours, consistent with the exhaustion of essential precursors and energy sources required for CA biosynthesis (Fig 7C). A complete carbon flux distribution, leading to CA biosynthesis during decay phase, is depicted in Fig 7D [36].

Fig 8 presents the metabolic flux profiles, over time, in _S. clavuligerus_, highlighting key metabolic features such as amino acid uptake pattern (Fig 8A–8D), glycolysis (Fig 8E–8K), TCA cycle (Fig 8L, 8M ), pentose phosphate pathway (Fig 8N, 8O) and nitrogen metabolism (Fig 8P–8R). Temporal analysis of amino acid metabolism revealed that _S. clavuligerus_ preferentially utilized specific amino acids depending on the growth phase. Glutamate (Glu-L), aspartate (Asp-L), and serine (Ser-L) were rapidly depleted during EEP and EP, reflecting their significant role in supporting cellular growth (Fig 8C). Proline (Pro-L), threonine (Thr-L), lysine (Lys-L), cysteine (Cys-L), leucine (Leu-L), tyrosine (Tyr-L), glycine (Gly-L), and valine (Val-L) displayed high consumption rates during early growth, followed by a sharp decline after 20 hours, indicating their role as major nitrogen and carbon sources (Fig 8A). Phenylalanine (Ph-L) and alanine (Ala-L) showed lower but

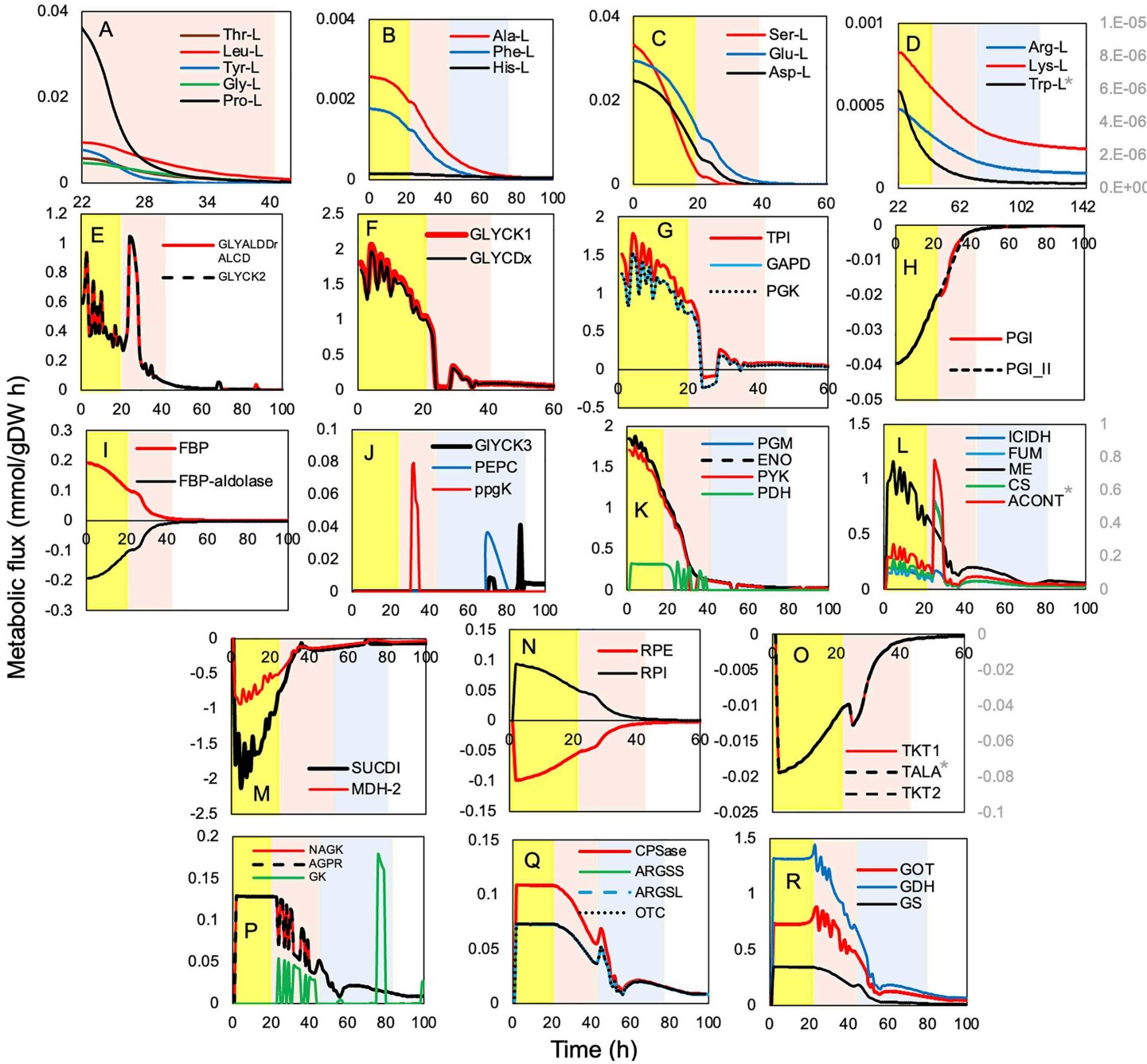

**Fig 8. Temporal metabolic flux profiles in *Streptomyces clavuligerus* during dFBA simulation.** Dynamics of amino acid metabolism **(A–D)**, glycolytic pathway **(E–K)**, tricarboxylic acid (TCA) cycle **(L–M)**, pentose phosphate pathway **(N–O)**, nitrogen metabolism **(P–R)**, and precursor metabolism for secondary metabolites. Early exponential (yellow), stationary (red), and decay (blue) phases are indicated. Enzymes: Transaldolase (TALA), transketolase (TKT1/2), glucose-6-phosphate isomerase (PGI), triose phosphate isomerase (TPI), ribose-5-phosphate isomerase (RPI), ribulose-5-phosphate epimerase (RPE), fructose-1,6-bisphosphate (FBP), aconitase (ACONT), isocitrate dehydrogenase (ICDH), fumarase (FUM), malic enzyme (ME), succinate dehydrogenase (SUCD1), malate dehydrogenase (MDH-2), glutamine synthetase (GS), glutamate dehydrogenase (GDH), glutamate oxaloacetate transaminase (GOT), carbamoyl phosphate synthetase (CPSase), and argininosuccinate synthetase (ARGSS).

sustained consumption over time (Fig 8B), suggesting they were metabolized consistently throughout growth. Histidine (His-L) (Fig 8B) and tryptophan (Trp-L) (Fig 8C) exhibited the lowest uptake rates and were primarily consumed during the stationary phase, implying a specific role during later stages of metabolism associated with secondary metabolite production [19]. The degradation of these amino acids produced intermediates such as α-ketoglutarate, succinyl-CoA, fumarate, oxaloacetate, pyruvate, acetyl-CoA, and acetoacetate, which contribute to the TCA cycle, amino acid degradation pathways, and secondary metabolism [37].

The glycolytic pathway and the pentose phosphate pathway (PPP), were highly active during EEP and EP, as indicated by the fluxes of glucose-6-phosphate isomerase (PGI) (Fig 8H) and triose phosphate isomerase (TPI) (Fig 8H), suggesting a significant generation of NADPH necessary for secondary metabolism. The non-oxidative branch of the PPP, including ribulose-5-phosphate epimerase (RPE), ribose-5-phosphate isomerase (RPI) (Fig 8N), transketolases (TKT1/2), and transaldolase (TALA) (Fig 8O), reached maximal fluxes between 0 and 30 hours, supplying ribose-5-phosphate and erythrose-4-phosphate for nucleotide and aromatic amino acid biosynthesis, respectively, with fluxes progressively declining as cultures entered the stationary phase [4,38]. Fluxes through glycolysis and PPP persisted during EP and SP, supporting cellular metabolism and precursor formation for secondary metabolite biosynthesis. $CO_2$ release via the PPP was observed and may contribute to the $CO_2$ production trends described above (see Fig 7B).

TCA cycle activity, including key enzymes such as isocitrate dehydrogenase (ICDH), fumarase (FUM), malic enzyme (ME), aconitase (ACONT), succinate dehydrogenase (SUCD1), and malate dehydrogenase (MDH-2), was high during EEP and EP, supporting ATP production and energy metabolism, but declined after 20 hours as cells transitioned into the stationary phase (Fig 8L, 8M). This decrease in TCA flux suggests a reduction in ATP and reducing equivalent production, with compensatory increases in PPP flux contributing to continued NADPH supply [11].

Nitrogen metabolism exhibited dynamic changes during growth (Fig 8P–8R). Glutamine synthetase (GS) and glutamate dehydrogenase (GDH) activity decreased under high nitrogen conditions, while carbamoyl phosphate synthetase (CPSase) and argininosuccinate synthetase (ARGSS) remained active, supporting the synthesis of key intermediates for CA production. During early growth, high flux through the urea cycle and glycolysis promoted the formation of CA precursors. The depletion of Glu-L around 48 hours limited these nitrogen pathways, constraining metabolite biosynthesis [37,39]. Phosphate limitation coincided with reduced growth rates and altered fluxes through nitrogen and carbon metabolism, influencing CA accumulation (Fig 8Q, 8R). Glutamate metabolism and its derivatives were fundamental contributors to CA biosynthesis throughout the culture period [40].

Integration of the carbon, nitrogen, and energy fluxes illustrates that during EEP and EP, high fluxes through glycolysis, PPP, TCA, and amino acid uptake supported active growth and energy generation. As the culture entered the stationary phase, amino acid consumption, TCA activity, and oxygen uptake decreased, whereas PPP flux remained partially active, sustaining precursor formation for secondary metabolism. CA excretion peaked during early SP and gradually declined, reflecting the temporal dynamics of precursor availability and metabolic flux distribution.

## Discussion

The GLYCAS-5 medium provides a nutrient-rich environment that directs cellular metabolism toward biomass generation rather than CA production, a pattern also observed in other *Streptomyces* species [1]. This indicates that the organism initially prioritizes growth before the onset of antibiotic synthesis. As commonly reported, antibiotics are produced as secondary metabolites during the late exponential (EP) or early stationary phase (SP), likely triggered by nutrient depletion and reduced growth rate (Fig 3) [41].

Based on physiological reasoning and previous studies, the rapid depletion of glutamate- and aspartate-family amino acids suggests their preferential utilization as combined carbon and nitrogen sources and reflects their metabolic linkage between nitrogen assimilation and the TCA cycle. This interpretation is consistent with the rapid biomass accumulation observed in GLYCAS-5 cultures (Fig 3).

Similarly, the more gradual consumption of pyruvate- and serine-family amino acids is consistent with their contribution to anabolic requirements during late exponential growth, as supported by model-based predictions of increased biosynthetic demand during this phase. Finally, the transient accumulation of leucine, cysteine, arginine, and glycine during early growth is consistent with literature reports describing peptide hydrolysis and temporary amino acid release associated with intracellular protein turnover [9,14].

As cultures entered the stationary phase, nearly all amino acids were depleted (Fig 4). This observation coincided with the transition from exponential growth to stationary phase. Such nutrient depletion is a known trigger for secondary metabolism in *S. clavuligerus* and suggests a metabolic shift from growth-associated pathways toward processes supporting clavulanic acid biosynthesis, in which amino acid-derived carbon skeletons may contribute to catabolic fluxes [14].

The unstructured kinetic model showed high agreement with experimental data for biomass, glycerol, clavulanic acid, and most amino acids, supporting its robustness in capturing the main growth and production dynamics. The lower fits obtained for ammonium and phosphate likely reflect physiological and regulatory processes not explicitly represented in the model, such as intracellular amino acid deamination and complex phosphate-mediated regulation of growth and clavulanic acid biosynthesis. In this context, the biomass-yield coefficients ($Y_{X/i}$) should be interpreted as effective parameters within a multi-substrate framework rather than as standalone physiological yields. Because glycerol provides most of the carbon and energy in GLYCAS-5 medium, small amino-acid uptake fluxes can be associated with substantial biomass formation, resulting in high apparent $Y_{X/i}$ values. Importantly, the relative ranking of these coefficients is consistent with experimental consumption patterns and known metabolic roles of glutamate- and aspartate-family amino acids in *Streptomyces*, supporting their use as quantitative descriptors of substrate contribution.

The topological analysis of the amino acid interaction network identified L-glutamate, L-aspartate, L-glycine, and L-cysteine as highly connected hub nodes (Fig 5). Independently, experimental concentration profiles showed that these amino acids were rapidly depleted during early growth (Fig 4). In contrast, transient increases were observed for certain amino acids, such as L-lysine and L-histidine, during the early exponential phase. Given the presence of casamino acids in the medium, this behavior is consistent with temporary peptide hydrolysis and release of free amino acids prior to their subsequent assimilation [9].

Together, these observations are consistent with the well-established central roles of glutamate-, aspartate-, and glycine-family amino acids in linking carbon and nitrogen metabolism and supporting biosynthetic demand during active growth. While the present analysis does not directly assess regulatory mechanisms, the combination of high network connectivity and rapid experimental depletion suggests that these amino acids occupy metabolically strategic positions within the network.

In this context, the network topology indicates that *S. clavuligerus* metabolism is structured around a limited set of multifunctional amino acids whose utilization is compatible with nitrogen-limited conditions associated with the transition toward clavulanic acid biosynthesis. Similar associations between amino acid centrality and secondary metabolite production have been reported in *Streptomyces lydicus*, where intracellular levels of glutamate and proline have been proposed as biomarkers of streptolydigin production [42].

Robustness analysis (RA) highlighted the differential effects of individual amino acids on clavulanic acid (CA) production and growth. L-arginine (Arg-L) and L-ornithine (Orn-L) were associated with high CA production while maintaining growth (Fig 6A), whereas L-glutamine (Gln-L) supported CA synthesis with a moderate reduction in growth rate. Amino acids from the α-ketoglutarate family, including L-proline (Pro-L) and L-glutamate (Glu-L), promoted high CA yields despite lower growth rates (Fig 6B). In contrast, Val-L, Ala-L, Lys-L, and His-L were less effective (Fig 6C), and inorganic nitrogen sources ($NH_4^+$ and urea) were suboptimal (Fig 6C). These trends are consistent with previous observations [1] and support the notion that specific amino acids are associated with distinct trade-offs between growth and secondary metabolism.

Notably, the limited effectiveness of lysine and histidine in promoting CA production, despite moderate predicted fluxes in the robustness analysis, may reflect regulatory constraints, slower assimilation kinetics, or transient intracellular accumulation that are not fully captured by the model. Acknowledging these discrepancies highlights current model limitations while preserving its value as a comparative framework for evaluating nitrogen sources and guiding hypothesis-driven optimization of fermentation conditions [40].

The role of amino acids as nitrogen donors and biosynthetic precursors for CA has been widely studied. Early reports demonstrated enhanced CA production with Arg and Orn supplementation [34,35,43,44]. Later studies explored Pro-, Lys-, Leu-, Glu-, Thr-, Trp-, Cys-, and Val-L [34], with Thr-L yielding the highest CA concentrations. Lynch and Yang (2004) showed that Lys plays an important role in CA biosynthesis: low Lys concentrations (1 g/L) did not induce CA formation, whereas adding degraded CA improved production (42 mg/L) compared to Lys supplementation alone (20 mg/L) [45]. Arg remains one of the most favorable precursors, while high Lys concentrations can repress CA synthesis via feedback inhibition. The pyruvate family (Leu, Ala, Val) shows variable effects, with Leu linked to secondary metabolism regulation in Streptomyces, although its specific role in CA synthesis is unclear [37].

Dynamic flux balance analysis (dFBA) simulations indicated that when both rapidly and slowly metabolized carbon sources are present, *S. clavuligerus* preferentially consumes the faster substrates before switching to the slower ones (Fig 7A). In the GLYCAS-5 medium, progressive nutrient depletion coincided with entry into the early stationary phase (EP), as reflected in glycerol and ammonium uptake profiles (Fig 7A, 7B). Clavulanic acid (CA) secretion initiated at approximately 40 h and reached a maximum near 70 h (Fig 7C), temporally aligning with the early stationary phase.

These dynamic patterns are consistent with those reported for other Streptomyces species [46]. For instance, in *S. xiamenensis* 318, amino acid concentrations peak during early growth and decline prior to maximum biomass accumulation, indicating nitrogen limitation during the late exponential phase [19]. More generally, nutrient limitation has been widely reported as a trigger for the metabolic transition from primary to secondary metabolism in *S. clavuligerus* and related species [47].

In this context, interpretation of the dFBA simulations supports a close association between CA production and macronutrient availability during batch culture. The coordinated depletion of carbon, nitrogen, and phosphate sources likely contributes to the transition from growth-associated metabolism to secondary metabolite biosynthesis (Fig 7C), in agreement with previous studies on the regulation of antibacterial and β-lactamase biosynthesis in [36,48]. This temporal coupling is consistent with the characteristic onset of secondary metabolism under nutrient-limited conditions [35,44].

At a finer metabolic scale, nitrogen metabolism has long been recognized as a key determinant of CA biosynthesis in S. clavuligerus. Previous studies have shown that excess nitrogen can repress CA production through nitrogen catabolite repression, whereas nitrogen limitation is associated with derepression of secondary metabolism [39]. In line with this framework, the temporal amino acid consumption profiles observed in Fig 8A–8D illustrate a shift from growth-associated nutrient utilization toward patterns coinciding with CA accumulation. Early and intensive consumption of nitrogen-rich amino acids such as glutamate, aspartate, and serine accompanies rapid biomass formation, whereas later utilization of aromatic and heterocyclic amino acids occurs concomitantly with CA production.

Consistent with these metabolic transitions, key enzymes involved in nitrogen metabolism, including glutamine synthetase (GS), glutamate dehydrogenase (GDH), and transaminases such as glutamate oxaloacetate transaminase (GOT) (Fig 8R), have been reported to be regulated by nitrogen availability in bacteria. Previous studies indicate that GS and GDH activities decrease under ammonium-rich conditions and increase under nitrogen limitation, with rapid GS inactivation upon ammonium addition followed by regulatory reactivation mechanisms such as adenylylation [40].

Further supporting this framework, the availability of organic nitrogen sources, particularly those associated with the Arg metabolic pathway, has been shown to increase intracellular amino acid levels and activate the CA biosynthetic gene cluster [37]. In addition, exogenous L-Glu has been reported to enhance antibiotic production, such as streptolydigin, highlighting its function as a biosynthetic precursor [40]. These results confirm the strong connection between

nitrogen metabolism—especially via L-Arg and L-Glu pathways—and the regulation of secondary metabolite production in *S. clavuligerus*. The metabolic flexibility of this organism enables it to adapt nitrogen use to environmental conditions, ensuring a balance between growth and antibiotic synthesis.

At a systems-level perspective, the evolution of $CO_2$ and $O_2$ fluxes (Fig 7B) reflects the metabolic demands during different phases of the batch culture. During early exponential and exponential phases, rapid glycerol and ammonium uptake sustain high oxygen consumption and $CO_2$ evolution (see Fig 7B), indicative of strong respiratory activity and biomass formation. As nutrients become depleted, the system shifts to lower oxygen uptake and $CO_2$ release, marking the transition toward secondary metabolism and CA production. In parallel, during the early exponential phase, the high uptake of carbon and nitrogen substrates is accompanied by increased activity in the Krebs cycle, as evidenced by the fluxes of isocitrate dehydrogenase (IDH), malate dehydrogenase (MDH), and succinate dehydrogenase (SUCD) (Fig 8L, 8M). This high metabolic activity coincides with elevated oxygen consumption and $CO_2$ emission, indicating intense respiratory activity and, therefore, enhanced antibiotic biosynthesis. Research efforts should focus on metabolic engineering approaches that include gene manipulation and targeted enzyme optimization to enhance the efficiency of amino acid utilization. Moreover, controlled feeding strategies could optimize CA production by modulating the availability of these key metabolites [1,37].

Consistent with these central metabolic shifts, the accumulation of fructose-1,6-bisphosphate (FBP) further implicates glycolytic regulation in CA production (Fig 8I). FBP acts as a flux-controlling intermediate and may signal the metabolic switch linking carbon metabolism to secondary metabolism, consistent with the changes observed in central carbon fluxes (Fig 8E–8K). Alterations in carbon, nitrogen, and phosphorus fluxes within primary metabolism strongly affect secondary metabolite formation. As reported by Wentzel et al. (2012), PPP flux increases during secondary metabolism to meet NADPH demands, while reduced tricarboxylic acid (TCA) cycle activity limits NADPH generation [49].

The temporal activation of key enzymes in the pentose phosphate pathway (PPP)—transaldolase (TALA), transketolases (TKT1/2), ribose-5-phosphate isomerase (RPI), and ribulose-5-phosphate epimerase (RPE) (Fig 8N, 8O)—was maximal during early growth (0–30 h), reflecting high demand for NADPH and ribose-5-phosphate required for nucleotide and amino acid biosynthesis. Later, the decline in PPP activity paralleled the shift from energy generation to the production of reducing equivalents for CA biosynthesis [4].

The observed decrease in TCA cycle enzyme activities—including isocitrate dehydrogenase (ICDH), fumarase (FUM), malate dehydrogenase (MDH), and aconitase (ACONT)—supports the hypothesis that metabolic precursors are diverted toward CA production rather than growth (Fig 8L, 8M). This behavior is characteristic of the metabolic switch from primary to secondary metabolism, where CA synthesis is favored as nutrients become limiting. Similar findings have been reported for *S. coelicolor*, where upper TCA cycle enzymes remain active during secondary metabolism [49]. In *S. clavuligerus*, succinyl-CoA availability appears to be a key regulatory factor, influenced by the activity of enzymes such as succinate dehydrogenase (SUCD1) and MDH-2. This highlights the interconnectedness of primary metabolism and secondary metabolite biosynthesis in *Streptomyces* [4].

Overall, the complex regulatory networks governing carbon, nitrogen, and phosphorus metabolism converge to control CA biosynthesis. Understanding these interactions provides a foundation for future studies aimed at elucidating the molecular mechanisms driving this metabolic transition and for developing strategies to enhance CA yield.

## Supporting information

**S1 File. *Streptomyces clavuligerus* unstructured kinetic model construction.**
(PDF)

**S2 File. *In silico* analyses of *Streptomyces clavuligerus* iLT1021 genome-scale metabolic model.**
(XLSX)

**S3 File. Correlation between the kinetic model and dynamic flux balance analysis (dFBA) simulations using the genome-scale metabolic model iLT1021 of *Streptomyces clavuligerus.***
(PDF)

## Acknowledgments

We acknowledge the support of the Bioprocess Research Group at the University of Antioquia for facilities and equipment essential to the experimental work.

## Author contributions

**Conceptualization:** Leon F. Toro-Navarro, Rigoberto Rios-Estepa.

**Formal analysis:** Leon F. Toro-Navarro, Laura Pinilla-Mendoza.

**Funding acquisition:** Leon F. Toro-Navarro.

**Investigation:** Laura Pinilla-Mendoza, Rigoberto Rios-Estepa.

**Methodology:** Rigoberto Rios-Estepa.

**Project administration:** Leon F. Toro-Navarro.

**Writing – original draft:** Leon F. Toro-Navarro, Laura Pinilla-Mendoza, Rigoberto Rios-Estepa.

**Writing – review & editing:** Leon F. Toro-Navarro, Laura Pinilla-Mendoza, Rigoberto Rios-Estepa.

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
