## [Decision Letter · Decision Letter 0]

2 Dec 2025

Dear Dr. Toro-Navarro,

We look forward to receiving your revised manuscript.

Kind regards,

Vasu D. Appanna

Academic Editor

PLOS ONE

Journal Requirements:

2. We are unable to open your Supporting Information file S2_file.sif. Please kindly revise as necessary and re-upload.

Reviewers' comments:

Reviewer's Responses to Questions

**Comments to the Author**

1. Is the manuscript technically sound, and do the data support the conclusions?

Reviewer #1: Yes

Reviewer #2: Yes

2. Has the statistical analysis been performed appropriately and rigorously?

Reviewer #1: Yes

Reviewer #2: Yes

3. Have the authors made all data underlying the findings in their manuscript fully available?

Reviewer #1: Yes

Reviewer #2: Yes

4. Is the manuscript presented in an intelligible fashion and written in standard English?

Reviewer #1: Yes

Reviewer #2: Yes

Reviewer #1: The manuscript is scientifically sound, clearly written, and the modeling and analyses are applied appropriately. My comments focus on improving clarity, mainly separating experimental results from interpretation, explaining unusually high yield coefficients, and adding brief methodological detail to ensure reproducibility. These are minor revisions and do not affect the validity or significance of the work.

Reviewer #2: In this manuscript, the authors use experimental evidence in tandem with in silico analysis to model the growth of streptomyces clavuligerus in complex medium. Given its importance in producing the beta-lactamase inhibitor clavulanic acid, the authors suggest the data within can be used to generate genetically engineered bacteria to better generate this antibiotic.

The paper is sound. I have a few minor recommendations:

- Figure captions should include more details (n value and statistical analysis if applicable)

- Figure three is missing the x axis legend (Time in h)

- If the resolution on figure 5 can be improved, it would increase legibility

- Remove the image of a mitochondrion in Figure 7, which are not present in bacteria and thus misleading

**Do you want your identity to be public for this peer review?** For information about this choice, including consent withdrawal, please see our Privacy Policy

Reviewer #1: No

Reviewer #2: No

---

## [Author Response · Author response to Decision Letter 1]

4 Jan 2026

Dear Academic Editor and Reviewers,

We greatly appreciate the positive assessment of the scientific soundness of the study and the constructive suggestions provided. All comments have been carefully addressed, resulting in substantial improvements in methodological clarity, figure presentation, and separation of experimental observations from interpretation. Below, we provide a detailed, point-by-point response. All changes have been incorporated into the revised manuscript and highlighted in the tracked-changes version.

Data Availability Statement

The Data Availability Statement has been updated to explicitly indicate that the amino acid interaction network reconstructed from the iLT1021 genome-scale metabolic model is publicly available in the Network Data Exchange (NDEx) repository under accession ID 4b9fe51c-e7f1-11f0-a218-005056ae3c32, in full compliance with PLOS ONE data availability requirements. All other relevant data are provided within the manuscript and Supporting Information files.

We believe that all reviewer comments have been fully addressed and that the revised manuscript meets the standards and scope of PLOS ONE. We appreciate the time and effort invested by the editor and reviewers and hope that the revised version will be suitable for publication.

Sincerely,

Leon Felipe

---

## [Decision Letter · Decision Letter 1]

18 Jan 2026

Dynamic metabolic modeling of Streptomyces clavuligerus in complex medium highlights nutrient-dependent metabolic transitions associated with clavulanic acid biosynthesis

PONE-D-25-57065R1

Dear Dr. Toro-Navarro,

We’re pleased to inform you that your manuscript has been judged scientifically suitable for publication and will be formally accepted for publication once it meets all outstanding technical requirements.

Kind regards,

Vasu D. Appanna

Academic Editor

PLOS One

Additional Editor Comments (optional):

Reviewers' comments:

Reviewer's Responses to Questions

**Comments to the Author**

Reviewer #1: All comments have been addressed

Reviewer #2: All comments have been addressed

2. Is the manuscript technically sound, and do the data support the conclusions?

Reviewer #1: Yes

Reviewer #2: Yes

3. Has the statistical analysis been performed appropriately and rigorously?

Reviewer #1: Yes

Reviewer #2: Yes

4. Have the authors made all data underlying the findings in their manuscript fully available?

Reviewer #1: Yes

Reviewer #2: Yes

5. Is the manuscript presented in an intelligible fashion and written in standard English?

Reviewer #1: Yes

Reviewer #2: Yes

Reviewer #1: All comments have been addressed. The revisions greatly enhance clarity and transparency, and no further changes are necessary from my side.

Reviewer #2: I have no further comments. The authors have responded to all my concerns. I believe the manuscript to be acceptable.

**Do you want your identity to be public for this peer review?** For information about this choice, including consent withdrawal, please see our Privacy Policy

Reviewer #1: No

Reviewer #2: No

---

## [Editor Report · Acceptance letter]

PONE-D-25-57065R1

PLOS One

Dear Dr. Toro-Navarro,

I'm pleased to inform you that your manuscript has been deemed suitable for publication in PLOS One. Congratulations! Your manuscript is now being handed over to our production team.

Kind regards,

on behalf of

Dr. Vasu D. Appanna

Academic Editor

PLOS One